# Unsupervised Cross-Task Generalization via Retrieval Augmentation

**Bill Yuchen Lin**[†]    **Kangmin Tan**[†]    **Chris Miller**[†]    **Beiwen Tian**[‡]    **Xiang Ren**[†]
[†] University of Southern California      [‡] Tsinghua University
{yuchen.lin,kangmint,millercs,xiangren}@usc.edu

## Abstract

Humans can perform unseen tasks by recalling relevant skills acquired previously and then generalizing them to the target tasks, even if there is no supervision at all. In this paper, we aim to improve this kind of cross-task generalization ability of massive multi-task language models, such as T0 and FLAN, in an unsupervised setting. We propose a retrieval-augmentation method named ReCross that takes a few *unlabelled* examples as queries to retrieve a small subset of upstream data and uses them to update the multi-task model for better generalization. ReCross is a straightforward yet effective retrieval method that combines both efficient dense retrieval and effective pair-wise reranking. Our results and analysis show that it significantly outperforms both non-retrieval methods and other baseline methods. [1]

## 1  Introduction

Advances in pre-training techniques for large language models (LMs) have considerably improved natural language processing (NLP) models on various important tasks via fine-tuning with labeled data. While these fine-tuned models are impressive in their target tasks, they can hardly generalize to unseen tasks. This thus makes it difficult to approach the general linguistic intelligence that we ultimately want an NLP model to enjoy. A promising avenue is to train a massively multi-task model that learns a large set of NLP tasks. However, in real-world applications, users often expect a multi-task NLP model can also perform unseen tasks that they are interested in. These users may only be able to provide a few *unlabeled* examples (i.e., the input-only data) of the target tasks with natural-language instructions. How can we generalize the multi-task model to unseen tasks without labels? This desirable ability is dubbed "unsupervised cross-task generalization."

Recent studies show that multi-task prompted training makes language models better in cross-task generalization, especially when natural-language instructions are used for formatting the training data (Ye et al., 2021; Sanh et al., 2021; Wei et al., 2021). The general recipe is to first fine-tune a text-to-text language model such as T5 (Raffel et al., 2020) on a multi-task mixture of diverse NLP datasets that are converted to sequence-to-sequence formats. We use the term *upstream learning* to refer to this multi-task training stage. Given a target task that is unseen during upstream learning, we want the upstream multi-task model to also perform well on it via reusing the previously acquired knowledge. FLAN (Wei et al., 2021) and T0 (Sanh et al., 2021) both use natural language (NL) instructions as prompts to reformat the data of various NLP tasks for upstream learning and generalization. Their results suggest that NL instructions are keys to unsupervised cross-task generalization.

Despite of the exciting results from Wei et al. (2021) and Sanh et al. (2021), their studies are limited to the analysis of the task generalization performance of the *frozen, target-agnostic* upstream models (i.e., FLAN and T0). We argue that the generalization performance can be further improved if we can exploit the unlabeled data of target tasks as hints for adjusting the upstream model to a more

---

[1]Our data, code, and supplementary materials are at https://inklab.usc.edu/ReCross/.

36th Conference on Neural Information Processing Systems (NeurIPS 2022).

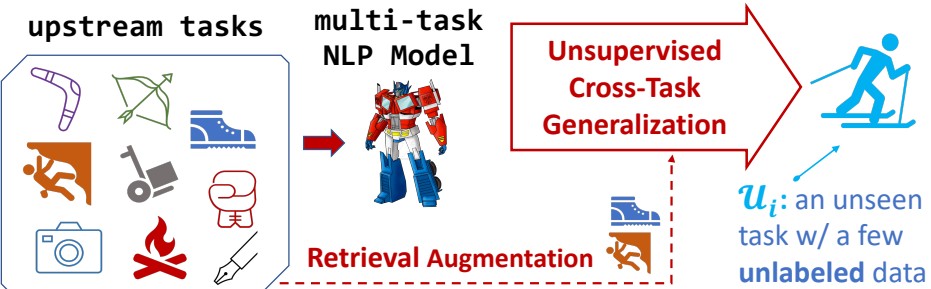

Figure 1: **The unsupervised cross-task generalization problem.** In the upstream training stage, we train a multi-task NLP model, $\mathcal{M}$, with a diverse collection of upstream tasks. In the generalization stage, given an unseen task $\mathcal{U}_i$ with a few unlabeled examples $Q_i$, we want to update the upstream model (via retrieval augmentation) such that it can generalize to the target task.

dedicated, target-aware model. Intuitively, the upstream examples that share similar skills with the target task should help the task generalization if the upstream model could recap these skills via retrieving. Motivated by this idea, we propose to further improve the cross-task generalization ability of upstream models via *retrieval augmentation* from the upstream data.

The key challenge of such retrieval augmentation is to predict the *example-level utility* for cross-task generalization, which we introduce with details in Sec. 2. To address the challenges, we present a two-stage retrieval-augmentation framework, ReCross, for unsupervised cross-task generalization in Section 3. Specifically, we pre-compute a dense index by encoding all upstream data as dense vectors. Given a set of unlabeled examples, we first use them to retrieve an initial set of upstream data by using encoded queries to efficiently search over the dense index. Then, we apply the reranking module to carefully analyze the utility of each candidate example. To get such a reranker, we fine-tune a cross-encoder model with distant supervision mined by a novel algorithm. Finally, we take top-ranking retrieved data to fine-tune the upstream model for a few steps and use this updated model for inference on the target task in the future (i.e., the retrieval augmentation and model update is a one-time procedure for each unseen task).

To more efficiently evaluate generalization methods without losing the generality, we train a variant of T0-like models, named BART0, which has comparable performance with T0-3B yet is 8x smaller. Our extensive experiments show that the proposed ReCross outperforms the baseline methods by a large margin. For example, ReCross improves the non-retrieval methods by 4 points on the overall performance of 10 target tasks and similarly on a few BigBench tasks. We also analyze the distribution of the retrieved data to understand the behavior of retrieval-augmentation methods better and find that ReCross has a very different distribution compared to semantic retrieval baselines.

## 2   Problem Formulation

**Massively Multi-Task Language Models.**   To build a general NLP model that can serve a wide range of real-world downstream applications, it is important to train a massively multi-task upstream model. We assume there are $N$ different **upstream tasks** (e.g., sentiment analysis of IMDB reviews), dubbed as $\{\mathcal{T}_1, \ldots, \mathcal{T}_N\}$. We use $D$ to denote the collection of all labeled data for these upstream tasks (i.e., the **upstream data**), which are then used for training a massive multi-task model $\mathcal{M}$ (e.g., BART, T5, and other Transformer-based models). The datasets of these upstream tasks are all converted to a shared *text-to-text* format using natural-language instruction templates such as PromptSource (Bach et al., 2022) to reformat data of different NLP tasks. This pipeline has become a common approach, adopted by several recent massive multi-task models for NLP, such as T0 (Sanh et al., 2021), FLAN (Wei et al., 2021), and CrossFit (Ye et al., 2021).

**Unsupervised Cross-Task Generalization.**   In real-world scenarios, it is very common that users to want a general multi-task model to perform tasks of their interest, even if their target tasks are never seen before by the upstream model. For these unseen target tasks, users usually can provide only a few *unlabeled* examples (i.e., the input-only data) of them for specifying the task instructions. This is the reason why we need to study how to generalize a multi-task LM to unseen tasks with only a few

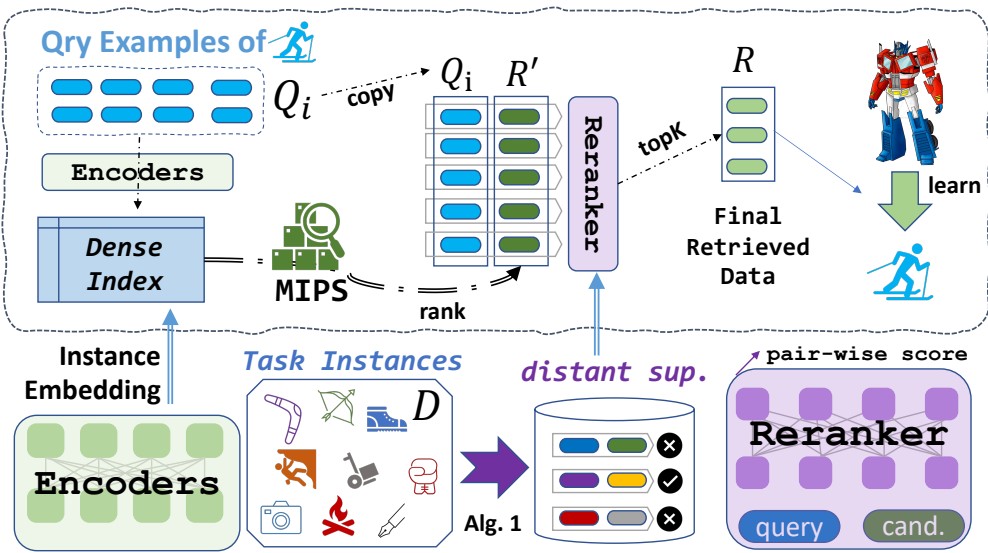

Figure 2: **ReCross is a retrieval-augmentation method for unsupervised cross-task generalization.** We reuse the encoder layers of the upstream model (green) to build a dense index, which consists of vectors of the upstream examples $D$. We also propose an algorithm to generate distant supervision for training a reranker, which takes a pair of examples as input and outputs a score. During the evaluation, we encode query examples $Q_i$ for querying the index to get initial ranking results $R'$, and then pair them with the queries again for reranking. Finally, we take the top-K results (i.e., $R$) for generalizing the upstream model $\mathcal{M}$ to the unseen task $\mathcal{U}_i$.

*unlabeled* examples, i.e., *unsupervised cross-task generalization*. For instance, in Fig. 1, the *unseen task* $\mathcal{U}_i$ is a coreference-resolution task that is not covered by the upstream training (the top-right box in Fig. 1). We have only a few inputs for it as the "*hints*" for cross-task generalization, which we call query examples $Q_i$. Our objective is to use the query examples $Q_i$ to enhance the performance of upstream model $\mathcal{M}$ on the unseen task $\mathcal{U}_i$. For evaluating such unsupervised cross-task generalization methods, we test the enhanced model with a held-out labeled data of each target task.

**Challenges.** Standard fine-tuning approaches (with or without meta-learning designs) for few-shot cross-task generalization are not feasible here. We have to adjust the upstream model based on only a few *input-only* examples for the unseen task. Intuitively, upstream examples that share similar skills with the target task $\mathcal{U}_i$ should be more beneficial than other upstream data. Thus, one naive idea is to first estimate the utility of each upstream example for $\mathcal{U}_i$ and then re-train a dedicated model $\mathcal{M}_i$ via a weighted learning method (e.g., examples of more utility are trained with larger loss).

However, such a target-aware weighted re-training method cannot scale, because the upstream data is usually very large and there can be a large number of unseen tasks from users in real-world applications. In addition, it is particularly challenging to estimate the utility scores of upstream data for a given unseen task, as we do not have ground-truth annotations for learning this. Although there are some existing studies on task-to-task relatedness and transferability (Vu et al., 2020; Lange et al., 2021; Padmakumar et al., 2022), most of them are not designed for unsupervised settings and few are done with multi-task (prompted) upstream models. Moreover, these prior analyses are mainly limited to the task-level analysis and they may not directly generalize to studying example-level utility, which is particularly important for the problem setup of this work.

## 3 ReCross: Retrieval Augmentation for Cross-Task Generalization

### 3.1 Overview

To address the above challenges for unsupervised cross-task generalization, we propose a retrieval-augmentation method named ReCross. The ReCross method is also based on the simple idea that we should exploit the upstream examples that enjoy better utility for a given unseen target task. Instead

of costly re-training from scratch, our method first retrieves a small subset of the upstream data for each unseen task. It then uses them to efficiently fine-tune the upstream model such that the updated model is generalized. This can ensure scalability to a great extent and benefit upstream models from re-learning target-specific acquired knowledge for cross-task generalization.

Ideally, we aim to retrieve the upstream examples that are the most beneficial ones for generalizing the upstream model towards a particular unseen task — ranking the upstream data by their example-level utility. To achieve this goal while preserving the efficiency, we first use the query examples to retrieve initial candidates via efficient maximum inner product search (MIPS) over a dense index, which consists of embedding vectors of all upstream examples (Section 3.2).

Based on the candidates from dense retrieval, we learn a reranking module for further improving the retrieval results (Section 3.3). The reranker is based on the cross-encoder architecture that takes a query-candidate pair of examples and outputs a more curated score of utility. Recall that we do not have any annotation for such example-level utility scores, and the only allowed resources are the upstream data and model. Therefore, we propose an algorithm to mine distant supervision from the upstream data for learning the reranker (Section 3.4). The overview of ReCross is shown in Fig. 2.

## 3.2 Dense Retrieval

To efficiently estimate the example-level utility for generalization, we propose to first employ a dense retrieval module that ensures high scalability. Specifically, we build a matrix $\mathbf{D} \in \mathbb{R}^{|D| \times d}$, where each upstream example in $D$ is encoded with a dense vector. Based on this dense index, we can now estimate the utility of an upstream example with its cosine distances to the encoded query examples in $Q$. That is to say, the upstream examples that are the nearest neighbors of query examples, are more likely to be beneficial for generalizing the upstream model $\mathcal{M}$ to the unseen target task.

To retrieve the candidate set $R'$, we use MIPS to search for the top-$K$ examples for each query example in $Q$, so $K = \lceil |R'|/|Q| \rceil$. (We introduce the details and other aggregation strategies in Appendix.) This dense-retrieval process is very efficient as we pre-compute the upstream index and perform MIPS for querying the candidates over the index on-the-fly during the generalization stage. We use the FAISS library (Johnson et al., 2019) in our implementation.

**Instance embeddings.** The example encoder is a key component of the dense-retrieval pipeline. An ideal example encoder is supposed to represent the underlying skills behind an example such that we can use the distances in the result embedding space to estimate utility for cross-task generalization. As we do not have annotations of utility scores for training an encoder, one may want to use pre-trained sentence embedding models such as SentenceBERT (Reimers and Gurevych, 2019). Our empirical results show that such semantics-based encoders cannot lead to much improvement over random retrieval results. We think there are two reasons for this failure. First, the *semantic* similarities between examples are not suitable for estimating the utility for generalization. Second, the *external* encoding modules do not reflect the nature of the upstream model which we want to generalize.

To address these two issues, we propose to use the encoding layers of upstream model $\mathcal{M}$ for computing the example embeddings. Without loss of generality, let us assume $\mathcal{M}$ to be a *text-to-text* Transformer that has multiple layers for both encoders and decoders such as BART. We encode an example by first obtaining the hidden representation of each token at the last encoder layer (i.e., a sequence of token vectors), and then performing mean-pooling over them to get a single dense vector to represent this example. By doing this, the produced example embeddings reflect the internal features of the upstream model, which are more relevant to the "thinking process" of the upstream model for the examples instead of the shallow semantic information.

## 3.3 Reranking Module

**Weakness of the dense retrieval.** Although dense retrieval is very efficient thanks to the MIPS support, the retrieval performance is limited by its two major weakness. First, it is a dual-encoder architecture that encodes the candidate example and the query example *separately*, which ignores informative features behind token-to-token attention across a pair of examples. Second, it is too costly to frequently update the example encoder, which prevents us from learning to refine the retrieval results with distant supervision (if any). Therefore, we design a re-ranking stage where we train a cross-encoder to further enhance the dense-retrieval results with mined distant supervision (Sec. 3.4).

**Encoding query-candidate pairs.** The cross-encoder architecture has been widely used in sentence-pair classification tasks such as natural language inference and paraphrase detection. We here use a cross-encoder to encode the *concatenation* of a query example and a candidate example. Specifically, we fine-tune a RoBERTa (Liu et al., 2019) model to classify whether an example pair is a positive or negative match. The confidence of classifying such a pair to be positive can thus be used as the utility score of the candidate upstream example for this query example. On top of this, we then develop a reranking module for further improving retrieval performance as follows.

**Scoring paired data.** To re-rank the initially retrieved data by the dense retriever, we apply the cross-encoder on all pairs of query examples $Q$ and candidate retrieved examples $R'$, producing scores of all $|Q| * |R|'$ query-candidate pairs. For each candidate example $r \in R'$, we use the average of all cross-encoder scores involving $r$ as its utility score. Finally, we take the top-$K$ examples based on this new ranking of candidate examples in $R'$ as the final retrieved data $R$. We use *upsampling ratio* $\mu$ to denote the ratio between $R'$ and $R$, i.e., $\mu = |R'|/|R|$.

## 3.4 Mining Distant Supervision for Reranking

How do we train such a re-ranking module? Recall that we only have access to the upstream data $D$ and must not use any data from the unseen tasks at this stage. Inspired by meta-learning works, we propose an algorithm (Alg. 1) to mine distant supervision data for creating a *training-as-testing* environment for learning the reranker. Our key motivation is to examine the utility scores of candidate examples by assessing the generalization performance of updated models that are fine-tuned with these candidates as if we use them for real unseen tasks. Such more realistic estimation of utility scores can thus help us train a reranker to predict.

---

**Algorithm 1: Distant Supervision Creation**

**Input:** $\mathcal{M}$; $D$; $\mathcal{T}_q$
**Output**: $Z = (Z_q, Z_p, Z_n)$

---

$D_{\mathcal{T}_q} \leftarrow \{x \in D | x \text{ is an example of } \mathcal{T}_q\}$
$Z_q \leftarrow \text{Sample}(D_{\mathcal{T}_q}); \quad H_q \leftarrow \text{Sample}(D_{\mathcal{T}_q})$
$R_Z \leftarrow \text{DenseRetrieve}(Z_q, D)$
/* Delete retrieved examples from the same task as queries. */
$R_Z \leftarrow R_Z.\text{discard}(D_{\mathcal{T}_q})$
**foreach** *round* **do**
  $R_Z.\text{shuffle}()$
  /* Split retrieved examples into $n$ groups */
  $\{G_1, ..., G_n\} \leftarrow R_Z.\text{split}()$
  **foreach** $G_i$ *in* $\{G_1, ..., G_n\}$ **do**
    $\mathcal{M}' \leftarrow \mathcal{M}.\text{copy}()$
    $\mathcal{M}'.\text{fine\_tune}(G_i)$
    $\ell \leftarrow \mathcal{M}'.\text{calc\_loss}(H_q)$
    **foreach** $x \in G_i$ **do**
      $scores[x].\text{append}(\ell)$
      /* Score each in the group w/ the loss. */

/* Use mean group score as score for single examples */
**foreach** $x \in R_Z$ **do**
  $score[x] \leftarrow \text{mean}(scores[x])$
/* Sort $R_Z$ by $score$ in increasing order. */
$R_Z.\text{sort}(\text{key}: score, \text{order}: \text{increasing})$
$Z_p \leftarrow \text{First } W \text{ items of } R_Z$
$Z_n \leftarrow \text{Last } W \text{ items of } R_Z$

---

Specifically, we define a data point of such distant supervision as a tuple $Z = (Z_q, Z_p, Z_n)$: 1) $Z_q$ is a set of query examples of a particular task $\mathcal{T}_q$; 2) $Z_p$ is the set of positive examples from other tasks; 3) $Z_n$ is the set of negative examples from other tasks. We expect that $Z_p$ is of more utility for generalization than $Z_n$ if $Z_q$ would be a query set for the target task $\mathcal{T}_q$. To this end, we first randomly sample an upstream task $\mathcal{T}_q$ and use a small subset of its training data as the $Z_q$. Here, we also sample a larger held-out set $H_q$ examples of task $\mathcal{T}_q$ to facilitate utility estimation. Then, we apply the dense retriever using $Z_q$ as the query examples and get the retrieval results $R_Z$. This $R_Z$ is thus the candidate pool where we create $Z_p$ and $Z_n$. That is, $Z_p \subset R_Z$ and $Z_n \subset R_Z$. We discard examples that are from the $\mathcal{T}_q$, so that the generated tuples are closer to the real scenarios where we use the reranker on the query sets of unseen tasks.

Our criteria to select $Z_p$ and $Z_n$ from $R_Z$ is motivated by the hypothesis that a more suitable set of retrieved examples should improve the performance $\mathcal{M}$ on $\mathcal{T}_i$ after fine-tuning with it. Therefore, we iteratively sample a small subset from $R_Z$, then fine-tune $\mathcal{M}$ with it, and finally, use the fine-tuned model to evaluate on $Z'_q$. The performance of such a temporarily fine-tuned model can be seen as the utility score—how well this subset can help generalize $\mathcal{M}$ to the unseen task $\mathcal{T}_q$. Through multiple rounds of such sample-train-test procedures, we can thus score each example in $R_Z$ by taking the average of all test results where it is involved. With such a new ranking of examples in $R_Z$, we take the best $W$ examples as $Z_p$ and the worst $W$ as $Z_n$.

With such distant supervision, we then can create pair of query-positive instances and query-negative instances via pairing $Z_q$ with $Z_p$ and $Z_n$ respectively. Now we can fine-tune a RoBERTa-base model

by concatenating each pair and learning a binary-classification objective. The output logits of this trained model will be used for the reranking procedure as shown in Sec. 3.3.

## 3.5 Re-learning via Fine-Tuning with Retrieved Data

When we have the final retrieved data $R_i$ for a certain query set $Q_i$, we can now enhance the upstream model $\mathcal{M}$ for the unseen task $\mathcal{U}_i$. We use a small learning rate to continually fine-tune $\mathcal{M}$ with the retrieved upstream examples $R_i$ for a small number of steps. We find that the learning rate has to be very small so that this step can be seen as a natural continuation of the finished upstream training and avoid overfitting the retrieved data. We acknowledge that there could be more effective methods to reuse the query examples $Q$ as guidance for fine-tuning, and we leave this as future work. Please find more discussion on the hyper-parameter selection and configuration in our appendix.

# 4 Evaluation

In this section, we first introduce the experimental setups, including the task distribution, upstream learning details, and the configurations of the main experiments. We present experimental results and reveal some non-trivial findings with extensive analysis that justify the effectiveness of ReCross.

## 4.1 Evaluating Unsupervised Cross-Task Generalization

We follow Sanh et al. (2021) to use the templates from PromptSource (Bach et al., 2022) for converting data of different types of NLP tasks to text-to-text formats. In total, we have 36 upstream tasks and 10 target unseen tasks for our main experiments. The upstream tasks are the same as the ones that the T0 models used for upstream learning. We follow the evaluation protocol proposed by Sanh et al. (2021) and select the target tasks that are significantly different from the upstream tasks. Besides, we also include 5 additional tasks from the BIG-bench project (Srivastava et al., 2022) to create an even more out-of-distribution set of unseen tasks for analysis.

**Metric.** When we apply the natural-language templates for the test examples, we only keep the templates that can be evaluated with an exact match (classification, question answering, answer selection, etc.) so that it is feasible to use exact-match for evaluating all tasks. To allow a smoother grading, our metric also counts the cases when outputs and truths are sub-strings of each other, which we call **SoftEM**. The only difference between SoftEM and the standard EM is that it also counts the sub-string matches. We observe that sometimes even though T0-like models (including ours) answer the input questions correctly, their raw outputs are not exactly the same as the truth outputs generated by the PromptSource templates. In particular, the ground-truth outputs for multiple-choice QA tasks are often in the form of "[A/B/C/D]: [answer]", while the models often only output the id of the correct choice (e.g., "A") or the text of the answer. We also find that the model can output some noise (such as additional punctuation) after the answer (e.g., "True" vs "True."). The standard EM will discard such matches and cause inaccurate measurements. Although SoftEM might add false positives due to substring matches, we found it is very rare according to our manual inspection of the 10 tasks. Therefore, we choose to use SoftEM for a more precise evaluation. We report the results with the standard EM in Table 7 that also supports our findings.

## 4.2 BART0: Upstream Learning with a Smaller LM

The T0(pp) models are all very huge, and the smallest version, T0-3B (3 billion parameters), is still too large to be fine-tuned on popular affordable GPUs. We need a parameter-efficient alternative that makes the study on cross-task generalization more accessible to a broader community while keeping the generality. Thus, we fine-tune a BART-large (Lewis et al., 2020a) (0.4 billion parameters) following the recipe of training T0. Specifically, we sample 50k examples at most from each upstream task to build a large upstream dataset consisting of 1.7 million examples (i.e., $|D| = 1.7\text{m}$), and then we fine-tune a BART-large with 22k steps with this upstream dataset. Finally, we use the fine-tuned checkpoint as our upstream model $\mathcal{M}$ and name it **BART0**. Surprisingly, we find that BART0 and T0-3B have comparable zero-shot performance on the unseen target tasks, even though T0-3B is about 8x larger than BART0. More implementation details are shown in Appendix.

## 4.3 Setup and Configurations

In our main experiments, we use $|Q_i| = 16$ query examples for each unseen task $\mathcal{U}_i$ and retrieve $|R_i| = 512$ examples for augmenting BART0. In the fine-tuning stage, we use a learning rate of 1e-6 and a

| Target Task | T0-3B | **BART0** | Random | SBERT | ReCross[†] | **ReCross** | Δ |
|---|---|---|---|---|---|---|---|
| anli_r3 | 26.00 | 30.50 | $35.34_{\pm 1.52}$ | $32.64_{\pm 2.53}$ | $36.70_{\pm 0.53}$ | $35.76_{\pm 0.90}$ | 5.26 |
| h-swag | 34.40 | 39.40 | $33.84_{\pm 5.59}$ | $30.92_{\pm 7.82}$ | $44.36_{\pm 3.07}$ | $47.28_{\pm 2.95}$ | 7.88 |
| cb | 53.93 | 39.64 | $47.07_{\pm 1.25}$ | $48.00_{\pm 3.28}$ | $44.50_{\pm 4.20}$ | $44.79_{\pm 3.36}$ | 5.15 |
| wic | 45.70 | 46.70 | $41.04_{\pm 2.18}$ | $46.78_{\pm 2.22}$ | $49.90_{\pm 0.50}$ | $50.58_{\pm 0.24}$ | 3.88 |
| wsc | 50.00 | 57.88 | $52.50_{\pm 2.29}$ | $52.69_{\pm 6.13}$ | $59.27_{\pm 1.96}$ | $61.46_{\pm 1.47}$ | 3.58 |
| winogrande | 47.60 | 51.10 | $52.68_{\pm 0.83}$ | $52.18_{\pm 3.20}$ | $54.60_{\pm 1.35}$ | $55.46_{\pm 0.88}$ | 4.36 |
| arc-chan. | 41.30 | 35.70 | $33.28_{\pm 1.50}$ | $37.90_{\pm 1.22}$ | $37.78_{\pm 0.73}$ | $38.44_{\pm 0.99}$ | 2.74 |
| obqa | 38.50 | 34.40 | $28.72_{\pm 2.46}$ | $33.28_{\pm 1.24}$ | $36.98_{\pm 1.55}$ | $39.58_{\pm 2.80}$ | 5.18 |
| piqa | 45.30 | 36.10 | $37.00_{\pm 2.71}$ | $38.54_{\pm 2.17}$ | $41.34_{\pm 1.75}$ | $41.42_{\pm 1.02}$ | 5.32 |
| squadv2 | 30.60 | 32.40 | $29.86_{\pm 5.46}$ | $29.46_{\pm 0.84}$ | $30.26_{\pm 1.54}$ | $30.58_{\pm 1.61}$ | -1.82 |
| All@mean | 41.33 | 40.38 | $39.13_{\pm 2.06}$ | $40.24_{\pm 1.61}$ | $43.57_{\pm 0.68}$ | $44.53_{\pm 0.42}$ | 4.15 |
| @median | 41.33 | 40.38 | 39.93 | 40.91 | 43.43 | 44.31 | 3.93 |
| @min | 41.33 | 40.38 | 35.66 | 38.28 | 42.65 | 44.16 | 3.77 |
| @max | 41.33 | 40.38 | 40.59 | 41.76 | 44.51 | 45.07 | 4.69 |

Table 1: **The main experimental results (%) for unsupervised cross-task generalization in SoftEM.** Each result in the upper section is the average (and the std) performance of using 5 different query sets for a task. The lower section of this table reports the mean, max, min, and median of the overall performance (i.e., the average performance on all tasks) of these five rounds.

batch size of 4 to continually fine-tune all layers of BART0 for 2 epochs. As for re-ranking, we set the upsampling ratio $\mu = 2$, meaning that we first retrieve 1024 examples for reranking and use the top 512 ones as the final retrieved data. To obtain more convincing evaluation results, we average the scores of all target tasks to show the general zero-shot performance. For each task $\mathcal{U}_i$, we use five different query sets, $\{Q_i^{(1)}, \ldots, Q_i^{(5)}\}$, to conduct **five individual rounds of retrieval**, thus resulting in five average scores for all tasks. To get a comprehensive assessment, we report the mean, std, median, min, and max of these five overall scores in the lower part of Table 1. We present an ablation study on hyper-parameter configurations in Table 3 and include more details in Appendix.

## 4.4 Experimental Results

**BART0 vs T0-3B.** As mentioned earlier, we find that BART0 is comparable with the much larger T0-3B in terms of their zero-shot performance on our unseen tasks (41.33 vs 40.38). As we use BART0 as our base model for testing different retrieval-augmentation methods, its overall performance *40.38* is what we want retrieval-augmentation methods to beat. Note that when using BART0 and T0-3B for non-retrieval zero-shot inference, they do not use any information from the query examples, so their mean, median, min, and max are always the same.

**Random Retrieval.** The *Random* column shows the results when we randomly sample $R_i$ from the upstream data $D$ without using any information from $Q_i$. .

**SBERT and ReCross[†].** We use SentenceBERT (SBERT) as a strong baseline method to create a dense index of the upstream data, compared with our proposed indexing method, ReCross[†] (i.e., ReCross without reranking). We can see that ReCross[†] always outperforms the other methods. Even its minimum performance in the five rounds (*42.65*) is better than the maximum of the SBERT (*41.76*). Besides, the standard deviation also becomes much smaller (1.61→ 0.68), which means that improvement by the ReCross[†] is more consistent under different query sets.

The SBERT indexing relies mainly on the *semantic similarities* between a query example and the upstream data. Instead, our proposed ReCross[†] uses the hidden representations inside the upstream model $\mathcal{M}$ for representing examples. We believe using such an indexing method can better help us find examples that share *similar reasoning skills* acquired by the upstream model.

**ReCross = ReCross[†] + Reranking.** The full version of our ReCross with reranking can further improve the performance substantially on multiple dimensions. Both all@mean and median are improved by 1 point from the ReCross[†], and the std is also reduced from 0.68 to 0.42. The last column (Δ) in Table 1 shows its improvement compared to the base model BART0, and we can see that ReCross consistently outperforms non-retrieval methods (e.g., BART0) by a significant gap.

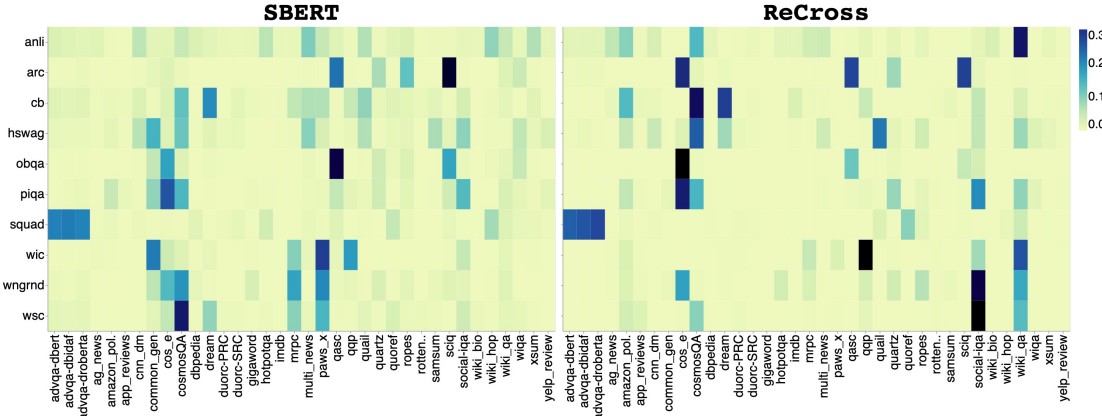

Figure 3: **The mapping between unseen tasks (as rows) and upstream tasks (as columns).** The darker upstream tasks take more percentage in retrieved data. For example, for the task `WIC`, ReCross retrieves a plurality of examples from `QQP` (about 30% of the retrieved examples).

To explore the potential benefits of retrieval-augmentation methods such as our ReCross, we also conduct the same experiments on five tasks selected from the BIG-Bench project. The results are shown in Table 2, where we can see that ReCross still outperforms the non-retrieval methods. An interesting case is the `movie_dialog` task, where the prompt in the template requires a model to output "same" or "different." However, both T0-3B and BART0 fail to follow the prompt instruction, and can only output "yes/no." Only when we use retrieval-augmentation methods, there are performance improvement on this task.

### 4.5 Analysis & More Findings.

**More configurations.** We have used a particular configuration in our main experiments that are in Table 1, which is $|Q|$=16, $|R|$=512, and $|u|$=2. In Table 3, we explore more configurations as ablation studies. The "Main Exp." row refers to the results shown in Table 1, and the configurations of other rows are only changed with one factor at a time. Even using a single query example, ReCross is better than BART0. However, when increasing the query size to 32,

| Task | T0-3B | BART0 | ReCross |
|---|---|---|---|
| hindu_knowledge | 24.75 | 23.48 | $24.87_{\pm 0.27}$ |
| known_unknowns | 47.83 | 43.48 | $47.17_{\pm 1.65}$ |
| logic_grid_puzzle | 23.60 | 20.70 | $17.12_{\pm 6.29}$ |
| strategyqa | 47.70 | 48.30 | $49.76_{\pm 0.80}$ |
| movie_dialog | 0.00 | 4.40 | $37.22_{\pm 13.26}$ |
| All@Mean | 28.78 | 28.07 | $35.23_{\pm 2.85}$ |

Table 2: **Results on a subset of BigBench tasks.**

we find that the performance starts to decrease, meaning that there could be an optimal query size for a certain $|R|$=512. We find that increasing $|R|$ is generally beneficial, while the all@mean decreases when $|R|$ is changed from 512 to 1024, although the max and the median slightly increased. Finally, we see that increasing $\mu$ increases the std. and does not improve the overall performance.

**Retrieved data distribution.** Figure 3 presents the difference between the methods in terms of their retrieved data. We draw the distribution of the retrieved data among different upstream tasks for each unseen task individually. From the heatmap, we can see that ReCross tends to have more dominant retrieved tasks (i.e., darker cells), while SBERT's results are more sparse. They both can identify that `squad` is most similar to the `adversarial_qa` tasks. Their behaviors are very different too. Taking the unseen task `winogrande` (wngrnd) as an example, we can

| Setup\All@ | Mean | std. | Min | Max | Median |
|---|---|---|---|---|---|
| Main Exp. | 44.53 | 0.42 | 44.16 | 45.07 | 44.31 |
| $|Q|$=1 | 43.20 | 0.83 | 42.58 | 44.58 | 42.88 |
| $|Q|$=8 | 43.67 | 0.90 | 42.09 | 44.32 | 43.90 |
| $|Q|$=32 | 42.52 | 1.17 | 40.52 | 43.40 | 42.96 |
| $|R|$=256 | 40.80 | 0.83 | 39.45 | 41.68 | 40.96 |
| $|R|$=1024 | 44.02 | 1.43 | 42.26 | 45.35 | 44.59 |
| $\mu$=3 | 43.92 | 0.58 | 43.08 | 44.57 | 43.89 |
| $\mu$=4 | 43.91 | 0.99 | 42.76 | 45.10 | 44.26 |

Table 3: **The ablation study of ReCross.**

see that the SBERT retrieves from multiple upstream tasks such as `paws-x` and `cosmosQA`, but the ReCross mainly retrieves from `social-iqa`, `wiki-qa`, and `cos-e`. The experimental results in Table 1 show that ReCross produces a better performance than SBERT (i.e., 55.46 vs 52.18), while

it is not clear how we can predict such task correlation in advance. This suggests that we should explore more about the utility of instances and tasks in future work.

**More analysis.** In the appendix, we further presented some analysis to help understand "how" and "when" the retrieval augmentation works: Table 4, Table 5, Appendix A.1 A.2, and Appendix B. We investigate whether the utility of upstream examples in retrieval augmentation is related to the similarity in terms of the task formats. From Appendix A.1, we found some counterintuitive results. For example, if removing MCQA upstream tasks from the upstream index, then the ARC target task can have an even better performance, although it is an MCQA-formatted task. Thus, we hypothesize that similarity in terms of reasoning types is more important than format similarity for retrieval augmentation. After all, the upstream model has been already trained to work with these basic task formats. Re-learning the tasks of the same format might lead the model to overfit the seen domains. Additionally, to provide a more concrete analysis, we also present case studies with two specific tasks (CB and SQUADv2) in Appendix B.

Moreover, we conjecture the natural language instructions in the templates are necessary for ReCross to get impressive results. Therefore, we investigated two ways of perturbing the instructions and monitoring the performance changes in Appendix A.2. We find it is indeed true that perturbations of the instructions will lead to much worse performance. We believe that a rigorous, principled way of analyzing the correlation between query and retrieval examples will be a great future direction, given the strong evidence that ReCross works so well as such a simple method.

## 5 More Discussion

### 5.1 Practicality of unsupervised setting.

**Cost of obtaining task labels** The unsupervised setting in the paper does not require any human annotation of labels. For some tasks (NLG tasks in particular, e.g., summarization), the expected output (label) are open-ended and possibly lengthy and thus human annotation is much more expensive and time-consuming. Also, few-shot learning must ask humans to label examples for each new task, and it is thus less practical when there are a large number of emerging tasks from the users. Meanwhile, ReCross requires only a natural-language task template, which does not require potentially expensive manual annotation or domain expertise.

**Scalability & Real-Time response** Deploying the ReCross pipeline is a one-time process. All we need to do is to pre-compute the upstream index with LM and configure the reranker (a simple masked LM) by running our script. In production, once the users input the examples with NL instructions, we do not need to wait for any human annotations anymore, so it is much more efficient in the long run. In the scenarios where users only provide one query example and want to get its label from the model, ReCross also shows great performance (i.e., |Q|=1 in Table 1). It is then impractical to assume there are a few labeled data from the users too in such cases.

### 5.2 Empirical studies

The unsupervised ReCross performance is comparable to few-shot learning with label annotations. In Appendix D.2, we report the performance of directly fine-tuning BART0 with the labeled query examples. Although it is an unfair comparison with our previous ReCross results, we found that they are comparable. More importantly, the ReCross framework does not conflict with the few-shot setting. Given a labeled query set for a target task, retrieved examples from the ReCross can still improve few-shot learning as additional training data. We designed two simple methods for applying ReCross under the few-shot setting and report the empirical results in Appendix D.2. It turns out that ReCross can also boost the performance under the few-shot setting by about 3 points.

## 6 Related Work

**Multi-task training for task generalization.** Text-to-text Transformer language models such as T5 enable us to train a multi-task NLP model with a more straightforward recipe: mixing the data of multiple tasks into a unified seq2seq format, and then fine-tuning text-to-text LMs for implicit multi-task learning. UnifiedQA (Khashabi et al., 2020) is among the first works in this direction.

Although it shows great generalization performance within QA tasks, it can hardly generalize to other NLP tasks. Recent works, such as CrossFit (Ye et al., 2021), ExT5 (Aribandi et al., 2022), FLAN (Wei et al., 2021), T0 (Sanh et al., 2021), and InstructGPT Ouyang et al. (2022) focus on how to generalize a massively multi-task model across task boundaries in a much broader context.

Particularly, in the CrossFit framework (Ye et al., 2021), cross-task generalization requires a small number of labeled instances of the target task for fine-tuning. It is because the templates of CrossFit use the task names as the hard prefixes. Therefore, it is necessary to fine-tune the upstream model with a few examples that have the target task names as prefixes (i.e., few-shot learning), but this largely limits the application scenarios of these multi-task NLP models in practice. We instead focus on *unsupervised* cross-task generalization, where there is no *labeled* data of an unseen task (i.e., zero-shot learning). Using natural-language instructions as prompts, both FLAN and T0 show that it is promising to perform *zero-shot* cross-task generalization.

In this work, we also focus on such an unsupervised setting for cross-task generalization, while our problem setup is a bit different from the ones used in T0 and FLAN. As for the assumption about the unlabeled data, their setups can be seen as a special case of ours when $|Q| = 1$ for all unseen tasks. The evaluation protocols of T0 and FLAN assess the generalization performance of the upstream model as it is, and thus their evaluation is more about the quality of templates and the upstream training tricks. In contrast, our evaluation protocol can also study how to efficiently adjust the upstream model such that the updated models can generalize to new tasks without labeled data. Thus, we believe ours is a more general setup for studying unsupervised cross-task generalization.

**Retrieval augmentation in NLP.** We aim to retrieve useful examples from the upstream data and re-learning them for cross-task generalization. The proposed ReCross pipeline is inspired by open-ended QA methods such as DPR (Karpukhin et al., 2020), DrFact (Lin et al., 2021), and RAG (Lewis et al., 2020b). Retrieval augmentation also shows great performance in pre-training LMs (Guu et al., 2020). Besides, Wang et al. (2022) shows that learning with similar data via retrieval augmentation can improve the performance of a task-specific model. Rubin et al. (2022) show that retrieving better demonstration examples is also helpful for in-context few-shot learning of GPT-3 style language models (Brown et al., 2020). The key challenge in the problem setup of this work is to predict the utility of the examples for unseen tasks with the consideration of efficiency and scalability. We have discussed more details about this challenge and related works in Sec. 2.

# 7    Conclusion & Future Directions

We demonstrate that retrieval augmentation can largely improve the cross-task generalization ability to multitask LMs in unsupervised settings. Our proposed method, ReCross, is a straightforward yet effective retrieval method that combines both efficient dense retrieval and effective pair-wise reranking. Our empirical results show that it significantly outperforms both non-retrieval methods and other baseline methods. We perform ablation studies showing the impact of changing query sizes, retrieval sizes, upsampling ratios, etc. We also find the distribution of retrieved data for analyzing the behavior differences between ReCross and others. We believe that our paper will spur further research on retrieval-augmentation methods for cross-task generalization. Interesting future directions include: 1) improve the re-learning stage by including more information from query examples, 2) extend the distant supervision mining process as a self-training procedure, 3) rigorously analyze the correlation between upstream data and target tasks, etc.

# Acknowledgments

This research is supported in part by the Office of the Director of National Intelligence (ODNI), Intelligence Advanced Research Projects Activity (IARPA), via Contract No. 2019-19051600007, the DARPA MCS program under Contract No. N660011924033, the Defense Advanced Research Projects Agency with awards W911NF-19-20271, NSF IIS 2048211, and gift awards from Google, Amazon, JP Morgan, and Sony. We thank all collaborators in USC and the NeurIPS 2022 reviewers for their constructive feedback on the work.

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
