# Appendix (i.e., Supplementary Material) of "Unsupervised Cross-Task Generalization via Retrieval Augmentation" (Submission # 2811)

This appendix include more implementation details, additional experimental results for ablation studies, and more analysis as well as findings. Please note that we have uploaded our code too (named "ReCross" folder). We first further analyze the performance of ReCross with more ablation analysis in Sec. A, and present detailed case studies for specific datasets in Sec. B, and introduce more implementation details in Sec. C.

## A  Additional analysis

### A.1  Utility analysis by grouping upstream tasks.

Table 4 shows the results of ReCorss under the scenarios where one specific group of upstream tasks are excluded from the index. This allows us to evaluate the impact of various upstream task categories on each downstream task.

| Task | None | −MCQA | −SUM. | −EQA | −Stmt. | −CBQA | −S2txt | −TopCls | −ParaIden |
|------|------|-------|-------|------|--------|-------|--------|---------|-----------|
| ARC-c. | $38.44_{\pm0.99}$ | $39.36_{\pm0.86}$ | $37.94_{\pm1.51}$ | $39.54_{\pm1.24}$ | $37.94_{\pm1.51}$ | $39.32_{\pm0.54}$ | $37.32_{\pm1.77}$ | $37.94_{\pm1.51}$ | $37.94_{\pm1.51}$ |
| anli_r3 | $35.76_{\pm0.90}$ | $36.18_{\pm0.88}$ | $36.90_{\pm0.83}$ | $36.78_{\pm1.04}$ | $35.72_{\pm1.92}$ | $35.84_{\pm2.35}$ | $37.42_{\pm0.97}$ | $35.92_{\pm1.32}$ | $36.42_{\pm1.20}$ |
| hswag | $47.28_{\pm2.95}$ | $40.56_{\pm8.71}$ | $49.28_{\pm5.79}$ | $39.02_{\pm7.49}$ | $46.46_{\pm3.39}$ | $37.62_{\pm5.98}$ | $46.00_{\pm6.32}$ | $39.14_{\pm7.50}$ | $44.34_{\pm6.19}$ |
| obqa | $39.58_{\pm2.80}$ | $36.12_{\pm0.88}$ | $38.32_{\pm2.33}$ | $38.52_{\pm2.08}$ | $38.32_{\pm2.33}$ | $35.98_{\pm2.37}$ | $36.32_{\pm2.86}$ | $38.32_{\pm2.33}$ | $35.94_{\pm1.70}$ |
| piqa | $41.42_{\pm1.02}$ | $39.60_{\pm1.35}$ | $40.46_{\pm2.08}$ | $41.64_{\pm2.65}$ | $41.30_{\pm2.47}$ | $41.56_{\pm1.46}$ | $40.26_{\pm2.17}$ | $40.42_{\pm0.99}$ | $40.56_{\pm0.80}$ |
| squad2 | $30.58_{\pm1.61}$ | $31.70_{\pm2.02}$ | $31.64_{\pm1.63}$ | $33.10_{\pm2.48}$ | $30.70_{\pm1.61}$ | $31.06_{\pm1.91}$ | $30.70_{\pm1.61}$ | $31.60_{\pm1.90}$ | $30.70_{\pm1.61}$ |
| cb | $44.79_{\pm3.36}$ | $49.36_{\pm3.55}$ | $44.50_{\pm4.52}$ | $43.93_{\pm3.26}$ | $40.79_{\pm3.05}$ | $44.00_{\pm5.42}$ | $43.36_{\pm4.15}$ | $42.36_{\pm7.36}$ | $40.50_{\pm5.62}$ |
| wic | $50.58_{\pm0.24}$ | $49.82_{\pm1.12}$ | $49.96_{\pm0.93}$ | $50.08_{\pm0.96}$ | $48.96_{\pm2.47}$ | $48.90_{\pm2.16}$ | $50.30_{\pm0.79}$ | $49.74_{\pm0.73}$ | $49.42_{\pm0.92}$ |
| wsc | $61.46_{\pm1.47}$ | $58.04_{\pm2.78}$ | $60.23_{\pm2.66}$ | $60.54_{\pm1.23}$ | $58.85_{\pm3.67}$ | $59.19_{\pm2.47}$ | $59.69_{\pm2.21}$ | $60.19_{\pm1.45}$ | $59.54_{\pm3.27}$ |
| wngrnd | $55.46_{\pm0.88}$ | $53.30_{\pm1.52}$ | $52.34_{\pm3.94}$ | $51.00_{\pm4.94}$ | $54.44_{\pm3.12}$ | $53.82_{\pm2.59}$ | $52.20_{\pm5.32}$ | $52.20_{\pm3.33}$ | $50.74_{\pm3.96}$ |
| @mean | $44.53_{\pm0.42}$ | $43.40_{\pm0.92}$ | $44.16_{\pm0.47}$ | $43.41_{\pm1.20}$ | $43.35_{\pm0.89}$ | $42.73_{\pm0.75}$ | $43.36_{\pm1.08}$ | $42.78_{\pm1.38}$ | $42.61_{\pm0.96}$ |

Table 4: Performance on each downstream task when a given category of upstream tasks is removed from the upstream dataset and prevented from being retrieved. The column names are the task group names: MCQA=Multiple-Choice QA, SUM=Summarization, EQA=Extractive QA, Stmt.=Sentiment analysis, CBQA=closed-book QA, S2txt=structure-to-text, TopCls=Topic Classification, and ParaIden=Paraphrase Identification.

Our key findings are as follows:

- (1) Using all upstream tasks leads to the best overall performance, although for many target tasks there are some particular groups that are less useful than others. The last row shows this result and the summarization is the least useful group of upstream tasks.

- (2) The potential best performance of retrieval-augmentation methods can be even higher. That is, if we have an enhanced version of ReCross that can avoid examples from less useful groups, then the final performance can be even higher. For example, if ReCross were able to ignore MCQA examples for ARC task during retrieval augmentation, then the overall performance of ReCross can be even higher.

- (3) The utility analysis via grouping upstream tasks by their original task formulations does not align with general intuition. For example, people may think that MCQA (multiple-choice QA) should be more useful than other groups for the task of ARC, which is also a multiple-choice QA dataset. However, removing MCQA doesn't hurt the performance of ARC. Instead, it actually improves the performance by 1 point. We argue that the example-based utility is of more importance for analysis.

### A.2  Template Perturbation

To investigate the importance of templates in retrieval quality, we investigated two methods of perturbing the templates of query examples $Q$: 1. Simply concatenate the elements in the raw data. 2. Change the words in the templates to random words to remove the semantic meaning. See figure 4 for an example. We than used these updated query examples and the same setup and configurations described in Section 4.3 to perform unsupervised cross-task generalization.

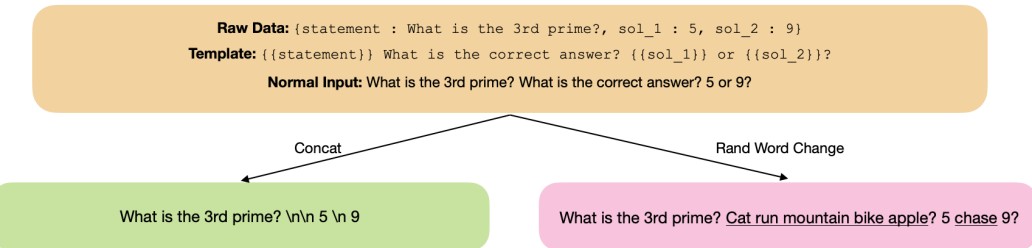

Figure 4: Example of concatenation and random word change perturbation.

| Target Task | T0-3B | BART0 | Random | SBERT | ReCross† | ReCross | Concat | Change |
|---|---|---|---|---|---|---|---|---|
| anli_r3 | 26.00 | 30.50 | $35.34_{\pm1.52}$ | $32.64_{\pm2.53}$ | $36.70_{\pm0.53}$ | $35.76_{\pm0.90}$ | $34.14_{\pm2.24}$ | $32.84_{\pm6.33}$ |
| h-swag | 34.40 | 39.40 | $33.84_{\pm5.59}$ | $30.92_{\pm7.82}$ | $44.36_{\pm3.07}$ | $47.28_{\pm2.95}$ | $35.74_{\pm5.06}$ | $35.40_{\pm10.82}$ |
| cb | 53.93 | 39.64 | $47.07_{\pm1.25}$ | $48.00_{\pm3.28}$ | $44.50_{\pm4.20}$ | $44.79_{\pm3.36}$ | $39.29_{\pm3.48}$ | $44.00_{\pm5.36}$ |
| wic | 45.70 | 46.70 | $41.04_{\pm2.18}$ | $46.78_{\pm2.22}$ | $49.90_{\pm0.50}$ | $50.58_{\pm0.24}$ | $46.88_{\pm2.93}$ | $47.32_{\pm1.91}$ |
| wsc | 50.00 | 57.88 | $52.50_{\pm2.29}$ | $52.69_{\pm6.13}$ | $59.27_{\pm1.96}$ | $61.46_{\pm1.47}$ | $52.31_{\pm5.17}$ | $57.31_{\pm1.75}$ |
| winogrande | 47.60 | 51.10 | $52.68_{\pm0.83}$ | $52.18_{\pm3.20}$ | $54.60_{\pm1.35}$ | $55.46_{\pm0.88}$ | $52.28_{\pm0.57}$ | $54.76_{\pm2.07}$ |
| arc-chan. | 41.30 | 35.70 | $33.28_{\pm1.50}$ | $37.90_{\pm1.22}$ | $37.78_{\pm0.73}$ | $38.44_{\pm0.99}$ | $37.92_{\pm0.48}$ | $38.24_{\pm1.20}$ |
| obqa | 38.50 | 34.40 | $28.72_{\pm2.46}$ | $33.28_{\pm1.24}$ | $36.98_{\pm1.55}$ | $39.58_{\pm2.80}$ | $36.12_{\pm3.14}$ | $38.56_{\pm2.06}$ |
| piqa | 45.30 | 36.10 | $37.00_{\pm2.71}$ | $38.54_{\pm2.17}$ | $41.34_{\pm1.75}$ | $41.42_{\pm1.02}$ | $39.76_{\pm0.99}$ | $42.16_{\pm1.86}$ |
| squadv2 | 30.60 | 32.40 | $29.86_{\pm5.46}$ | $29.46_{\pm0.84}$ | $30.26_{\pm1.54}$ | $30.58_{\pm1.61}$ | $30.74_{\pm1.66}$ | $30.10_{\pm1.22}$ |
| All@mean | 41.33 | 40.38 | $39.13_{\pm2.06}$ | $40.24_{\pm1.61}$ | $43.57_{\pm0.68}$ | $44.53_{\pm0.42}$ | $40.52_{\pm1.2}$ | $42.07_{\pm1.5}$ |
| @median | 41.33 | 40.38 | 39.93 | 40.91 | 43.43 | 44.31 | 40.96 | 41.69 |
| @min | 41.33 | 40.38 | 35.66 | 38.28 | 42.65 | 44.16 | 38.77 | 40.37 |
| @max | 41.33 | 40.38 | 40.59 | 41.76 | 44.51 | 45.07 | 41.61 | 44.33 |

Table 5: Two methods of template perturbation (concatenation and random word change) compared with main experiment results.

Table 5 shows that when we simply concatenate the elements in raw data, the performance degrades to a level close to random retrieval. On the other hand, if we construct the query examples as specified by the templates, even if we break the semantics of the template, the performance boost is largely preserved. This might mean that the formatting of input, for example the existence of parallel choices in some form, potentially plays an important role in the performance gain.

### A.3 Re-ranking for Random and SBERT

We evaluated training re-rankers for random and SentenceBERT retrievers. Specifically, we applied the same distant supervision mining methods introduced in Section 3.4 on data retrieved by **Random** and **SBERT**. Table 6 shows the results. We can see that reranking does not improve the results for both Random and SBERT retriever. We believe it is because that the initial retrieval results are not good enough, so that the distant supervision mined from them are thus also not of good quality.

| Target Task | Random | Random+RR | SBERT | SBERT+RR |
|---|---|---|---|---|
| anli_r3 | $35.34_{\pm1.52}$ | $31.58_{\pm4.39}$ | $32.64_{\pm2.53}$ | $28.10_{\pm4.83}$ |
| h-swag | $33.84_{\pm5.59}$ | $33.20_{\pm9.86}$ | $30.92_{\pm7.82}$ | $37.80_{\pm6.92}$ |
| cb | $47.07_{\pm1.25}$ | $40.71_{\pm1.84}$ | $48.00_{\pm3.28}$ | $40.86_{\pm7.80}$ |
| wic | $41.04_{\pm2.18}$ | $44.74_{\pm0.88}$ | $46.78_{\pm2.22}$ | $45.88_{\pm2.19}$ |
| wsc | $52.50_{\pm2.29}$ | $50.38_{\pm6.03}$ | $52.69_{\pm6.13}$ | $55.42_{\pm2.66}$ |
| winogrande | $52.68_{\pm0.83}$ | $49.44_{\pm13.80}$ | $52.18_{\pm3.20}$ | $53.02_{\pm3.49}$ |
| arc-chan. | $33.28_{\pm1.50}$ | $33.52_{\pm3.76}$ | $37.90_{\pm1.22}$ | $37.54_{\pm1.87}$ |
| obqa | $28.72_{\pm2.46}$ | $25.96_{\pm6.53}$ | $33.28_{\pm1.24}$ | $35.08_{\pm3.27}$ |
| piqa | $37.00_{\pm2.71}$ | $35.22_{\pm5.25}$ | $38.54_{\pm2.17}$ | $38.82_{\pm2.06}$ |
| squadv2 | $29.86_{\pm5.46}$ | $25.28_{\pm3.93}$ | $29.46_{\pm0.84}$ | $29.56_{\pm1.40}$ |
| All@mean | $39.13_{\pm2.06}$ | $37.00_{\pm2.91}$ | $40.24_{\pm1.61}$ | $40.21_{\pm1.83}$ |
| @median | 39.93 | 37.06 | 40.91 | 39.81 |
| @min | 35.66 | 33.32 | 38.28 | 38.45 |
| @max | 40.59 | 40.26 | 41.76 | 42.82 |

Table 6: Random and SBERT with Re-Ranking (RR) (bold font columns)

### A.4 Mining distant supervision for multiple iterations.

The algorithm that we proposed in Alg. 1 can be extended to an iterative process. That is, we can continually update the reranker module and uses the retrieved results from the latest reranker to mine the training data for the next iteration. Although this self-training style process sounds promising, our empirical results show that the overall performance starts to saturate after the first iteration and using the 2nd-iteration re-ranker won't improve the overall performance anymore. We think there can be better methods of continual learning to obtain a reranker module for better performance, while it is beyond the scope of this work. We hope this can be a promising future direction.

### A.5 Transferring ReCross for Larger Base Models.

Recall that we choose to use BART0 as our base model for its smaller size and comparable results. People may wonder what if we transfer the ReCross methods for larger base models. Therefore, we conduct a pilot study on this. Considering the size of T0, we choose to only fine-tune its last few layers of T0-3B and still use the prior materials from BART0 (i.e., the BART0-based index and the trained reranker). We found that the performance is not improved over simply using T0-3B for zero-shot inference. We conjecture there are two major reasons for this: 1) the parameter-efficient tuning method need to added here to improve the training efficiency, 2) the BART0-based index and the associated reranker do not align with the other models such as T0-3B. We admit this could be one limitation of our methods – i.e., the index and reranker are specific to the base model that is used to generate them. In order to address these challenges, we argue that studying the common space of the index created by different encoders will be an important direction.

## B Case studies

In this section, we discuss two specific datasets with detailed analysis as they have quite special results in Table 1 and Table 4.

### B.1 SuperGLUE CommitmentBank (cb)

For the SuperGLUE CommitmentBank dataset, instances retrieved by the BART retriever are predominantly multiple choice question-answering. However, heat map and remove-one-group analysis shows that re-training on instances from multiple choice question-answering seems to undermine the model's zero-shot performance on this dataset. We examined the output of the model and discovered that the model tends to make one type of error a lot more often when re-trained using multiple choice question-answering: instead of answering yes, no, possible, or impossible, it picks part of the discourse as its prediction.

For example:

Input: "Suppose A: I'm like, I'll get a job some day and my boss will pay for it, I'll be needed. B: Yeah. A: Because, um, I didn't want to go do it myself because I didn't think I was really going to use it. Can we infer that "he was really going to use it"? Yes, no, or maybe?"

Output: "A: I didn't want to go do it myself because I didn't think I was really going to use it."

We believe this is because the model misunderstood the people having the discourse (A and B) to be the options for answers. The abundance of the template of "A:xxx, B:xxx" in the SuperGLUE CommitmentBank dataset might be the reason why the BART retriever retrieved mostly from multiple choice question-answering in the first place.

### B.2 SQuAD V2

For the dataset SQuAD V2, the retriever typically finds upstream examples from extractive question answering datasets, which match the format of SQuAD V2 inputs closely. However, we find that when we exclude extractive question answering examples from the upstream dataset, performance on SQuAD V2 improves. To explain this unexpected result, we note that the majority of our test examples for SQuAD V2, despite being formatted as extractive question answering tasks, are examples which expect the model to output whether or not the question is answerable. The 'context and question' format of the SQuAD V2 examples causes the retriever to focus on extractive question answering examples, but because most of the examples focus on answerability (a distinct task from extractive question answering), these examples are not helpful.

We speculate that by excluding extractive question answering from the upstream dataset, the model avoids these misleading irrelevant examples and is able to retrieve more related examples for determining if a question is answerable. For example, our results show that when extractive question answering examples are excluded, the retriever finds examples from tasks such as Wiki QA, which asks whether or not a proposed answer is a valid answer to a given question (a more relevant task to determining if a question is answerable).

# C   Implementation details

## C.1   Retrieval aggregation.

Note that the target size of our retrieved data is $|R|$ and we have $|Q|$ query examples. To retrieve $|R|$ examples, we search for the top-$K$ examples for each query example, where $K = \lceil \frac{|R|}{|Q|} \rceil$, and then take the first $|R|$ of them when $K|Q| > |R|$. Our results have shown that this method is more effective than other strategies, such as combining the distance scores generated for each query example. Note that by retrieving the top-$K$ examples, we may repeat examples that are close to multiple query vectors. This effect is desirable because it allows us to naturally focus more on the especially relevant upstream examples in re-learning.

## C.2   Upstream learning.

**Upstream tasks.**   Here we refer to the T0's paper (cited in our main paper) for Figure **??**, which shows the list of upstream tasks and their categories. We use this taxonomy to conduct ablation study. Please find the link to download these datasets from huggingface/dataset from our submitted code. All datasets are publicly available and their license are suitable for open-source research. We do not see any ethical concerns from using such datasets for learning a model and developing the ReCross method to further improve their task generalization performance.

**Training details.**   We specify the hyper-parameters and the concrete for training the BART0 models in our submitted code. Please read the "Readme.md" file where we point to the script and configurations for training BART0. Our GPU type is Quadro RTX 6000 and 8000.

## C.3   Retrieval Methods.

Similarly, we leave the details such as the hyper-parameters and the concrete pipeline for running the retrieval augmentation methods (i.e., ReCross and the other baseline methods) in a unified framework that is presented in our code.

# D   Others

## D.1   Evaluation metrics.

**Results with the standard EM.**   In Table 7, we report our main experimental results (the equivalent results to those in Table 1) with the standard EM metric instead of the SoftEM metric used in Table 1. We can see that the relative performance from the ReCross framework is about the same as in Table 1, although the absolute numbers are mostly smaller due to a more strict matching by EM.

## D.2   Empirical results for few-shot learning.

We show the empirical results related to the few-shot setting in Table 8.

**Experimental setup.**   We assume that the labels of the examples in the query set are available, and directly use them to fine-tune the upstream model for learning the target task. We tune the hyper-parameters (epochs and learning rates) such that they do not overfit the few-shot data and lead to a better performance over BART0. Note that the real performance of few-shot learning performance may be lower than the ones in the table because there is not enough development data for us to tune hyper-parameters for each target task.

**Few-shot learning is not even better than the unsupervised ReCross.**   Although FS can outperform ReCross in some target tasks, the two approaches have very similar overall performance on 10 tasks. Even in such an unfair setting, ReCross shows great benefits to the users.

**ReCross and Few-Shot together can produce better performance.**   We attempted to use both the few-shot data and the retrieved data for generalization. The FS+RC(mix) method simply merge the 16 labeled query examples (i.e., few-shot) and the 512 retrieved data (by ReCross) to get a larger

| Target Task | T0-3B | BART0 | Random | SBERT | ReCross† | ReCross | Δ |
|---:|---:|---:|---:|---:|---:|---:|---:|
| anli_r3 | 24.30 | 24.30 | $27.80_{\pm2.12}$ | $25.62_{\pm2.35}$ | $31.02_{\pm0.87}$ | $30.18_{\pm1.48}$ | 5.88 |
| h-swag | 22.20 | 24.20 | $26.04_{\pm2.61}$ | $22.88_{\pm2.44}$ | $27.48_{\pm2.04}$ | $26.04_{\pm1.19}$ | 1.84 |
| cb | 49.29 | 26.79 | $31.64_{\pm3.27}$ | $34.21_{\pm5.12}$ | $30.00_{\pm2.65}$ | $31.57_{\pm6.18}$ | 4.79 |
| wic | 44.70 | 45.80 | $45.26_{\pm4.13}$ | $46.78_{\pm2.22}$ | $49.90_{\pm0.50}$ | $50.58_{\pm0.24}$ | 4.78 |
| wsc | 48.85 | 54.42 | $53.96_{\pm3.29}$ | $52.42_{\pm6.09}$ | $59.15_{\pm1.82}$ | $61.42_{\pm1.51}$ | 7.00 |
| winogrande | 47.00 | 49.50 | $50.44_{\pm0.57}$ | $50.80_{\pm2.89}$ | $54.16_{\pm1.18}$ | $54.42_{\pm1.10}$ | 4.92 |
| arc-chan. | 32.10 | 23.70 | $26.84_{\pm1.37}$ | $27.02_{\pm2.52}$ | $26.86_{\pm1.90}$ | $27.16_{\pm1.78}$ | 3.46 |
| obqa | 38.80 | 34.10 | $27.20_{\pm1.24}$ | $33.76_{\pm1.51}$ | $36.90_{\pm2.56}$ | $39.56_{\pm2.79}$ | 5.46 |
| piqa | 33.40 | 29.10 | $29.32_{\pm3.26}$ | $28.94_{\pm3.08}$ | $31.70_{\pm3.17}$ | $30.46_{\pm2.34}$ | 1.36 |
| squadv2 | 23.70 | 26.30 | $24.20_{\pm4.34}$ | $21.90_{\pm1.17}$ | $22.96_{\pm1.95}$ | $23.32_{\pm2.16}$ | -2.98 |
| All@mean | 36.43 | 33.82 | $34.27_{\pm1.66}$ | $34.43_{\pm1.14}$ | $37.01_{\pm0.94}$ | $37.47_{\pm0.73}$ | 3.65 |
| @median | 36.43 | 33.82 | 34.90 | 34.91 | 36.62 | 37.17 | 2.34 |
| @min | 36.43 | 33.82 | 31.33 | 32.91 | 36.22 | 36.93 | 1.05 |
| @max | 36.43 | 33.82 | 35.35 | 35.79 | 38.41 | 38.75 | 1.70 |

Table 7: **The main experimental results (%) for unsupervised cross-task generalization in the standard EM metric, i.e., the EM version of Table 1.**

| Target Task | BART0 | ReCross (ReX) | Few-Shot(FS) | FS+ReX(Mix) | FS+ReX(2-stage) |
|---:|---:|---:|---:|---:|---:|
| anli_r3 | 30.50 | $38.44_{\pm0.99}$ | $34.59_{\pm2.33}$ | $35.71_{\pm1.59}$ | $36.26_{\pm1.48}$ |
| h-swag | 39.40 | $35.76_{\pm0.90}$ | $42.61_{\pm2.15}$ | $44.04_{\pm3.60}$ | $43.99_{\pm1.92}$ |
| cb | 39.64 | $47.28_{\pm2.95}$ | $52.57_{\pm6.11}$ | $62.64_{\pm5.68}$ | $65.36_{\pm6.70}$ |
| wic | 46.70 | $39.58_{\pm2.80}$ | $48.22_{\pm2.10}$ | $49.23_{\pm1.52}$ | $48.21_{\pm2.57}$ |
| wsc | 57.88 | $41.42_{\pm1.02}$ | $53.15_{\pm3.80}$ | $55.65_{\pm7.82}$ | $54.54_{\pm5.22}$ |
| winogrande | 51.10 | $30.58_{\pm1.61}$ | $54.24_{\pm1.57}$ | $53.24_{\pm1.81}$ | $53.87_{\pm1.72}$ |
| arc-chan. | 35.70 | $44.79_{\pm3.36}$ | $36.36_{\pm2.20}$ | $36.34_{\pm2.64}$ | $37.50_{\pm2.94}$ |
| obqa | 34.40 | $50.58_{\pm0.24}$ | $34.49_{\pm4.21}$ | $38.45_{\pm2.68}$ | $37.15_{\pm2.63}$ |
| piqa | 36.10 | $61.46_{\pm1.47}$ | $47.38_{\pm4.58}$ | $51.93_{\pm2.72}$ | $52.08_{\pm1.95}$ |
| squadv2 | 32.40 | $55.46_{\pm0.88}$ | $41.92_{\pm6.68}$ | $51.30_{\pm3.23}$ | $50.38_{\pm6.46}$ |
| All@mean | 40.38 | 44.54 | 44.55 | 47.85 | 47.93 |

Table 8: **The few-shot related empirical results in SoftEM.**

dataset for fine-tuning BART0. The FS+RC(2-stage) method updates the model firstly with the 512 retrieved data and then train the fine-tuned model with the 16 FS data. Both methods show a great enhancement over FS and RC used separately. This is to say, RC is still beneficial in the FS setting.