# OpenReview forum: "Unsupervised Cross-Task Generalization via Retrieval Augmentation"
_NeurIPS.cc/2022/Conference — NeurIPS 2022 Accept_

### Official Review · Reviewer_NtmR · 2022-06-30

**Rating:** 6
**Confidence:** 3
**Soundness:** 3 good
**Presentation:** 3 good
**Contribution:** 2 fair

**Summary:**

This paper considers the problem unsupervised cross-task generalization of multi-task LMs such as T0, where only a few unlabeled examples from the target task are available. It proposes to retrieve, from the multi-task training corpus, samples that are likely to help the target task using the unlabeled target task inputs, and fine-tune the LM on these retrieved samples.
To retrieve helpful samples for a target task, they propose a retriever-ranker pipeline, where the more efficient retriever first retrieve, from the entire training set, an initial set of candidate samples, using the last-layer encoder representations from the upstream LM as embeddings for retrieval. It then trains a reranker using a meta-learning inspired approach, aiming to find training samples that are more likely to help an unseen target task.
They show that this retrieval-augmented approach achieves better cross-task generation than the original LM, as well as a naive retrieval approach based on semantic similarity (using SentenceBERT embeddings).

**Questions:**

Q1: Does this work assume the availability of prompt templates for the unseen target tasks? If I understand correctly, these templates are part of the "input" of the target task, which are assumed to exist? If this is correct, then it amplifies my main concern in the Weakness section. It is arguably more difficult / expensive to come up with these natural language prompts that work well for the upstream LM for each unseen task than to label a few samples.

**Strengths And Weaknesses:**

Strengths:

- The paper is clearly written. The problem setting and proposed approach is lucidly presented.

- Given that no target task labels are available in the "unsupervised" setting, the proposed method is sound, and produces non-trivial gains over the applying the vanilla upstream LM to unseen tasks.

Weaknesses:

- My main concern is the practicality or usefulness of this "unsupervised" setting. Given that only a few target task samples are required (the paper uses 16), the cost of obtaining the inputs vs. (inputs + labels) would probably not differ that much in the real world. It would at least be helpful to compare to the "supervised" cross-task generalization methods to show how much the gap is. And if this approach lags significantly behind a few-shot supervised one, it is arguably less expensive / time-consuming to simply label a few samples than to apply a complicated retriever-ranker pipeline.

EDIT after rebuttal: The added Appendix D.2 in the few-shot setting is welcome, and to some extent alleviates my concern here. Though it does not fully address this issue: Collecting labels is more expensive on certain tasks for sure, but this work only considers the tasks where the answer can be evaluated via exact match (e.g. multiple choice, classification, etc.). Arguably the cost for collecting 16 inputs vs. 16 (inputs + labels) does not differ that much. In addition, the concern over the availability of prompt templates for unseen tasks remains. Few-shot methods can transfer to new tasks with a few inputs + labels. I'm not convinced that this is inferior to the proposed paradigm of requiring a prompt template and a few inputs.

---

> ### Author Response · Authors · 2022-08-02
> **Comparisons with the few-shot setting, including new empirical results.**
>
> Thank you for your thorough review! We are pleased to hear that you feel that ReCross is technically solid. We understand the concern about the usefulness of the unsupervised setting (vs. the few-shot setting). We provide more clarification below and have done some additional experiments. Please see the new Appendix D.2 for the experiments and analysis.
>
> In this response, we list the reasons why we think our setting is practical, and show that ReCross can also boost the performance under the few-shot setting with new empirical results.
>
> ##  Practicality of unsupervised setting
>
> ### Cost of obtaining task labels
>
> The unsupervised setting in the paper does not require any human annotation of labels. For some tasks (NLG tasks in particular, e.g., summarization), the expected output (label) are open-ended and possibly lengthy and thus human annotation is much more expensive and time-consuming. Also, few-shot learning must ask humans to label examples for __each__ new task, and it is thus less practical when there are a large number of emerging tasks from the users. Meanwhile, ReCross requires only a natural-language task template, which does not require potentially expensive manual annotation or domain expertise.
>
> ### Scalability & Real-Time response
>
> Deploying the ReCross pipeline is a one-time process. All we need to do is to pre-compute the upstream index with LM and configure the reranker (a simple masked LM) by running our script. In production, once the users input the examples with NL instructions, we do not need to wait for any human annotations anymore, and thus it is much more efficient in the long run at scale.
>
> ### One query example at a time
> In the scenarios where users only provide one query example and want to get its label from the model, ReCross also shows great performance (i.e., |Q|=1 in Table 3). Now, users aim to get the labels from the model because they don’t know the truth. It is then impractical to assume there are a few labeled data from the users too.
>
> ## Emprical studies
>
> __The unsupervised ReCross performance is comparable to few-shot learning with label annotations.__  In Appendix D.2, we report the performance of directly fine-tuning BART0 with the labeled query examples. Although it is an unfair comparison with our previous ReCross results, we found that they are comparable. Plus, we find it is also very challenging and time-consuming to tune the hyper-parameters (because there is no dev set) and know when to stop to avoid overfitting. This again suggests that the unsupervised setting is more practical in production.
>
> __ReCross can boost few-shot performance.__ More importantly, the ReCross framework does not conflict with the few-shot setting. Given a labeled query set for a target task, retrieved examples from the ReCross can still benefit few-shot learning as additional training data. We designed two simple methods for applying ReCross under the FS setting and report the empirical results in Appendix D.2.  It turns out that ReCross can also boost the performance under the FS setting by about 3 points.
>
> All in all, we believe that the problem setting studied in this paper can be very common and less costly than the few-shot setting when there are a large number of users and emerging tasks. Plus, our method ReCross can also boost performance in the few-shot setting settings.
>
> ## Cost of creating prompt templates
>
> The prompt templates used in the paper are natural language (NL) instructions from the PromptSource repo, which are part of the input texts. They are the foundations of current unsupervised cross-task generalization methods (including T0, FLAN, instruction-GPT, and ours).  And it’s not that difficult for users to create a template for a target task. Instead, only when we have such NL elements, the users can naturally interact with the LMs by describing the task in their own words. For example, “select the best choice as the answer to the question: xxx..” or “Is the review supportive or not? Review: XXXX. Yes or No.”
>
> We can also consider problems such as summarization or tasks which rely on domain-specific knowledge as examples of tasks where forming a simple prompt template or gathering inputs is substantially easier than building an effective number of input/output pairs for few-shot learning. Although there may be some sensitivity to the chosen prompt, prior work (e.g., Scao and Rush 2021, “How Many Data Points is a Prompt Worth?”) suggests that prompt selection is not a dominant driver of performance.
>
> Without such NL templates, then we will need to either put the task names as the prefix of the input sequences or just give no signals for the LMs to distinguish/connect across tasks. Neither method can hardly enable cross-task generalization without any training data.  We also studied the importance of the NL templates in retrieval as an ablative study in Appendix A.2.

---

> > ### Author Response · Authors · 2022-08-08
> > **Follow-up discussion on the few-shot settings**
> >
> > Dear Reviewer NtmR,
> >
> > Thank you very much for reading our response and raising your rating for the ReCross paper. We just noticed there were a few additional comments ("EDIT after rebuttal") in the main review. To make the discussion clearer, we list four scenarios for the training and testing stages:
> >
> > 1. Upstream Training with Templates + Unsupervised Generalization (N inputs with templates)
> > 2. Upstream Training with Templates + Few-Shot Generalization (N inputs with templates + N labels)
> > 3. Upstream Training with Templates + Few-Shot Generalization (N inputs ***w/o*** templates + N labels)
> > 4. Upstream Training ***w/o*** Templates + Few-Shot Generalization (N inputs ***w/o*** templates + N labels)
> >
> > Setting 1 is our main experiment in the paper, and Setting 2 is what we add in Appendix D.2. We show that ReCross can improve the BART0's performance in both Setting 1 and Setting 2. For Setting 3, our preliminary results show that the performance of BART0 is much worse than in Setting 2 and we think it is because the training and generalization stage is inconsistent with the model. For Setting 4, the base model was even worse because it is less capable of using the cross-task information from the upstream data, and even adding meta-learning elements in the upstream learning cannot help much (we refer to the CrossFit paper for similar experiments).
> >
> > As for the cost of annotation cost, we agree that adding 16 annotation labels for one given task is possible, while the unsupervised setting is more general if we consider the scalability and the efficiency in the inference stage. Therefore, Setting 1 is less expensive than 2. Given that the base model performance in Setting 3/4 is not at the same level as in Setting 1/2, so we did not consider them in our evaluation. We will add a comprehensive analysis including these two settings as well to provide more discussion.
> >
> > Finally, we would like to argue again that the main focus of our paper is not on a new setting, but on the retrieval augmentation method (ReCross) that can help cross-task generalization. Therefore, our evaluation used the settings, which most prior works like T0, FLAN, and other recent instruction-based generalization methods focus on. We love the discussion with you and will definitely add new empirical results and new discussion in our final version of the paper.
> >
> > Thank you very much again!
> >
> > Best regards,
> >
> > Authors of the ReCross paper

---

> > > ### Comment · Reviewer_NtmR · 2022-08-09
> > > **Re: Additional Follow-up**
> > >
> > > Thanks for the follow-up. Looking forward to the additional results and discussion that you plan to add.

---

### Official Review · Reviewer_fTmN · 2022-07-10

**Rating:** 6
**Confidence:** 4
**Soundness:** 2 fair
**Presentation:** 2 fair
**Contribution:** 3 good

**Summary:**

This paper presents an approach called ReCross for fine-tuning an LLM by retrieving seen-task data that is particularly relevant to unlabelled inputs from an unseen task. The seen-task examples that are most relevant are retrieved from an index which contains representations from a model that has been fine-tuned on the seen-task data. Following the retrieval, a re-ranker is used to score each retrieved example-query pair, and the top K are used for fine-tuning the model before evaluating on the unseen target task. Experiments on a range of NLU tasks including classification, QA etc. show that ReCross outperforms instruction-tuning baselines.

**Questions:**

Please see the previous section which includes suggestions as well.

**Limitations:**

Missed the section discussing limitations, where is it included?

**Strengths And Weaknesses:**

Strengths:
This paper shows that carefully selecting data from related tasks for fine-tuning before zero-shot evaluations can help to improve performance. This is sort of a test-time adaptation of the model parameters at a coarser granularity than at an example-level. This means that the retrieval cannot be based on semantic similarity which is why the model used for creating the retrieval index has been fine-tuned on a large pre-selected subset of the seen-task data. The entire setup being explored is an important and high-impact problem to study, and retrieval augmentation provides a compelling solution worth exploring as done in this paper.
Also, the Bart0 vs. T0-3B result is quite interesting.

Weaknesses:
- The paper is missing a qualitative discussion on what sorts of examples are retrieved and what sorts of features might be in play for retrieving useful similar examples. Figure 3 doesn’t seem to be providing too many intuitions for this. This makes it a bit hard to understand where the improvements are coming from.
- The role of the reranker is unclear. Improvements over and above the simpler version of ReCross without the reranker seem small, mostly non-existent given the standard deviation, which is bizarre especially given how complex the whole distant supervision setup is. Would it be better to just remove this from the main paper and replace it with more analysis, data-related ablations and discussion on where the improvements are coming from? ReCross without the distant supervision is still interesting and perhaps a bit simpler.
- The conclusion from the random retrieval baseline that says “This suggests that it is promising to study better retrieval methods to get consistent improvement in overall performance” seems to only hold true for winogrande..maybe it isn't the right conclusion? Random retrieval is a control experiment to remove confounds and understand how to interpret the improvement from ReCross. E.g. h-swag, anli_r3, squadv2 do not benefit from ReCross.
- The paper, while comprehensive, can sometimes be verbose. Perhaps some revisions would help to pare it down. Figures 1 and 2, as is, may not be helping the reader to understand the approach. Is the example in Figure 1 an actual retrieval from the amazon polarity and gigawords tasks? These don’t seem like they should help, should they? Maybe cherry-picking a clearer example could help with this one? Figure 2 might benefit from being broken down into stages.

EDIT: Updating the score from 4-->6 after the author response.

---

> ### Author Response · Authors · 2022-08-02
> **More analysis; Importance of the reranker; Clarifications.**
>
> Thank you for your thorough and constructive review! We are pleased to hear that you feel that our ReCross is a compelling solution to an important and high-impact problem to study. We also appreciate your concerns and questions, each of which we address in detail below.
>
> ## More analysis
>
> In Appendix, we presented some analysis to help understand “how” and “when” the retrieval augmentation works: Figure 4, Table 5, Appendix A.1~A.2, and Appendix B.
>
> We investigate whether the utility of upstream examples in retrieval augmentation is related to the similarity in terms of the task formats.  From Appendix A.1, we found some counterintuitive results. For example, if removing MCQA upstream tasks from the upstream index, then the ARC target task can have an even better performance, although it is an MCQA-formatted task. Thus, we hypothesize that similarity in terms of reasoning types is more important than format similarity for retrieval augmentation. After all, the upstream model has been already trained to work with these basic task formats. Re-learning the tasks of the same format might lead the model to overfit the seen domains. Additionally, to provide a more concrete analysis, we also present case studies with two specific tasks (CB and SQUADv2) in Appendix B.
>
> We think the natural language instructions in the templates are necessary for ReCross to get impressive results.  Therefore, we investigated two ways of perturbing the instructions and monitoring the performance changes in Appendix A.2. We find it is indeed true that perturbations of the instructions will lead to much worse performance.
>
> We will move these analyses to the main paper once given more space limit in the final version.  We believe that a rigorous, principled way of analyzing the correlation between query and (optimal) retrieval examples will be a great future direction, given the strong evidence from this paper that ReCross works so well.
>
> ## The importance of the ReRanker.
>
> __Performance__: In Table 1, we can see that using the reranker not only improves the mean, min, and max but also reduces the std. Specifically,  the overall mean is improved by 1 point in SoftEM (i.e., about 2.3% relative improvement compared with no-reranking ReCross) -- two tasks get 7% relative improvement, four tasks get 2~4% gain, and three others get 1% gain. The min of the overall performance (when using multiple query sets) is improved by 4% relatively.  These results show that re-ranking can consistently enhance the overall performance and make the ReCross more stable.  Therefore, we argue that the role of the reranker is of great importance for ReCross.
>
> __Extensibility__: The reranker also provides a great space for future research. Indexing the upstream data can be very time-consuming so we don’t want to do that frequently. Therefore, if we have more training signals to improve the retriever, it is very important to have an efficient reranker module for learning to rank.
>
> ## Clarification regarding the conclusion on random retrieval
>
> Sorry for the confusion about this statement. We wanted to show the potential benefits of retrieval augmentation by comparing the BART0 column with the **maximum performance** of the Random retrieval baseline.  We have clarified this statement in the revised paper.
>
> The max performance among the five rounds of random retrieval is usually comparable to or larger than the performance of the vanilla BART0 for all tasks (looking at the 2nd column and the 3rd column’s mean+std in Table 1). For example, although the mean of random for SquadV2 is 29.86 which is smaller than BART0’s 32.40, the max of random is about 35.32.
>
> The better maximum performance of “lucky” rounds of random retrieval suggests that it is worth developing better retrieval augmentation methods. Plus, it also suggests that if given suitable retrieved data, such a simple “re-learning” method could already enhance the upstream model.
>
> ## Other suggestions.
> - Conciseness:  We rephrased a few paragraphs to make them more concise and they are highlighted in blue.
> - Figures: We refined Figures 1 and 2 accordingly based on the comments.
> - Limitations: We talked about the limitations of our work in Sec. 3.5 (for the re-learning method) and Sec. 6 -- the three future directions can be interpreted as the limitations.
>
> Thank you very much for the valuable questions and suggestions.

---

> > ### Comment · Reviewer_fTmN · 2022-08-09
> > **Update after rebuttal**
> >
> > Thanks for the updates and the detailed response.
> >
> > - The qualitative analysis seems interesting and, agreed, is probably sufficient for this paper.
> > - I hadn't noticed that the re-ranker helps to reduce the stdev in many cases, that seems like a useful outcome, worth staying in the main paper I suppose. It is still true that the gains from the re-ranker are just not very good or very consistent. The paper would just be a lot stronger if the vanilla recross method -- which provides better improvements over baselines and frankly, is a lot more elegant -- were the focus of the paper. And the re-ranker can be introduced as a secondary extension -- one of the signals you mentioned, that can help nudge scores to be higher sometimes. This mostly just needs some reframing; the base ReCross method without reranking is quite a nice approach and will probably draw majority of the interest. That being said, I'll certainly raise my score so the paper isn't blocked by this.
> > - The conclusion from the random retrieval baseline is still not quite right and calling the max score from the trials "lucky" seems really not right. Random retrieval is an important control to eliminate confounds and help determine the true value added by smarter retrieval. Reading too much into the result of a single trial seems gnarly and I'd suggest against it.

---

> > > ### Author Response · Authors · 2022-08-09
> > > **Thank you for the reply!**
> > >
> > > Thank you very much for your detailed reply and raised score!
> > >
> > > We will revise the final version accordingly based on these valuable suggestions and comments. Specifically, we will reframe the introduction of the reranker such that we have more space to add our analysis to the main paper. We will also rephrase the conclusion about the random retrieval such that the analysis will focus more on eliminating confounds instead.
> > >
> > > Thank you very much again for your thoughtful review and detailed discussion! :D

---

> ### Author Response · Authors · 2022-08-05
> **A friendly reminder to respond to author rebuttal (the ReCross paper)**
>
> Hi Reviewer fTmN,
>
> Thank you again for your review! Based on your thoughtful feedback, we wrote a detailed rebuttal covering the following points:
> 1. more analysis to understand the performance gains by ReCross
> 1. the importance of the role of reranking module
> 1. the clarification of our statement about the comparisons between random retrieval and BART0 model
> 1. our revisions according to your great editing suggestions
>
> In light of the imminent discussion deadline (Aug. 9), it would be awesome to know if our rebuttal sufficiently addressed your concerns and questions. If your concerns were well addressed, would you please consider raising the score for the ReCross paper? By the way, Reviewer NtmR has raised the rating to 6 (weak accept) after the rebuttal.
>
> We understand that you are busy, so we would appreciate it a lot! Thank you very much! :D
>
> Sincerely,
>
> Authors of the ReCross paper

---

> > ### Author Response · Authors · 2022-08-08
> > **Another friendly reminder (the ReCross paper)**
> >
> > Dear Reviewer fTmN for the ReCross paper,
> >
> > Thank you for your time and efforts again in reviewing our paper. We kindly remind you that the discussion period will end very soon. We believe that we sincerely and successfully address your comments by covering all the questions.
> >
> > If you have any further concerns or questions, please do not hesitate to let us know.
> >
> > We understand that you are very busy, so we would appreciate it a lot! Thank you very much! :D
> >
> > Sincerely,
> >
> > Authors of the ReCross paper

---

### Official Review · Reviewer_YUN8 · 2022-07-11

**Rating:** 6
**Confidence:** 4
**Soundness:** 3 good
**Presentation:** 3 good
**Contribution:** 3 good

**Summary:**

The authors present ReCross, a retrieval augmentation method to improve cross-task generalization of seq2seq models. Starting with a T5-like model ("upstream model") trained on a diverse set of tasks (in seq2seq format), the training data is then embedded using the encoder's top layer and stored in a dense index. To evaluate on an unseen task, the query is used to retrieve training data examples that might be helpful in evaluating this new example (using a two-stage retrieve+rerank process). These helpful examples are used to fine-tune the upstream model, and the model is finally used to perform the target task.

**Questions:**

* How does ReCross compare to the other methods of Table 1 if using exact match as the metric? I see it's implemented in the code, so I assume this was tested?

**Limitations:**

The authors adequately address in section 3.5 the fact that their continuous fine-tuning approach is quite simple.

**Strengths And Weaknesses:**

Strengths:
* This is a simple but effective method for allowing a model to retrieve helpful previously seen training data, and re-learn from it. As the authors say, there are probably approaches that would be more effective than continuous fine-tuning. However, it's impressive how well such a simple method already works.
* The authors provide a reasonable set of baselines and an informative choice of ablations (including those in the Appendix). We see the importance of choosing a way of embedding examples that is compatible with the upstream model's internal representation. We also see the contribution of the reranker to the overall performance, and the interactions of various central hyperparameters.

Weaknesses:
* ~One crucial issue I see with this work is that the authors create a new metric (SoftEM) and use it throughout the paper without properly motivating it. They refer to the Appendix, stating "We discuss more on this selection with illustrative examples in Appendix". No such discussion or examples are found in the Appendix, as far as I can tell. It's not clear whether the proposed method would still perform as well using a more standard metric.~ EDIT: Fixed in the latest draft

---

> ### Author Response · Authors · 2022-08-02
> **Empirical results with the standard EM; Motivation of the SoftEM.**
>
> Thank you very much for your comments and questions! We are pleased to hear that you feel that our ReCross is a simple but effective method of unsupervised cross-task generalization. We understand your concerns and questions. We have revised the paper accordingly and added the new empirical results in __Appendix D.1__.
>
> ## Results with the standard EM.
>
> Thank you for bringing this up! We add the EM-version of Table 1 in Appendix D.1  for a more comprehensive evaluation as the reviewer suggested (please check Table 7). The relative difference between the methods is similar to the ones with the SoftEM, and our findings still remain almost the same.
>
> ## Motivation of the SoftEM metric.
> In Line 233-234 of the initial submission, we describe the only difference between SoftEM and the standard EM is that SoftEM also counts the substring matches. We adopt this variant because we observed that sometimes even though T0-like models (including ours) answer the input questions correctly, their raw outputs are not exactly the same as the truth outputs generated by the PromptSource templates. In particular, the ground-truth outputs for multiple-choice QA tasks are often in the form of “[A/B/C/D]: [answer]”, while the models often only output the id of the correct choice (e.g., “A”) or the text part of the choice. We also find that the model can output some noise (such as additional punctuation) after the answer (e.g.,  “True” vs “True.”). The standard EM will discard such matches and cause inaccurate measurements. Although SoftEM might add false positives due to substring matches, we found it is very rare according to our manual inspection of the 10 tasks. Considering both SoftEM and EM have their pros and cons, we will present both results in our final version.

---

> > ### Comment · Reviewer_YUN8 · 2022-08-09
> > **Response to rebuttal**
> >
> > Thank you for clarifying the differences between SoftEM and EM, as well as for providing extra analyses in the Appendix.

---

> ### Author Response · Authors · 2022-08-05
> **A friendly reminder to respond to author rebuttal (the ReCross paper)**
>
> Hi Reviewer YUN8,
>
> Thank you again for your review! Based on your thoughtful feedback, we wrote a detailed rebuttal about the metrics. To make it easier for you to check out the EM-based results, we attach a brief version of the new Table 7 (in Appendix D.1) below in this comment.
>
> In light of the imminent discussion deadline (Aug. 9), it would be awesome to know if our rebuttal sufficiently addressed your concerns and questions. If your concerns were well addressed, would you please consider raising the score for the ReCross paper? By the way, Reviewer NtmR has raised the rating to 6 (weak accept) after the rebuttal.
>
> We understand that you are busy, so we would appreciate it a lot! Thank you very much! :D
>
> Sincerely,
>
> Authors of the ReCross paper
>
>
> | task             | T0_3B  | BART0  | Random | SBERT  | ReCross-init. | ReCross | \Delta |
> |------------------|--------|--------|--------|--------|---------------|---------|--------|
> | Overall - mean   | 36.43% | 33.82% | 34.27% | 34.43% | 37.01%        | 37.47%  | 3.65%  |
> | Overall - Median | 36.43% | 33.82% | 34.90% | 34.91% | 36.62%        | 37.17%  | 2.34%  |
> | Overall - Min    | 36.43% | 33.82% | 31.33% | 32.91% | 36.22%        | 36.93%  | 1.05%  |
> | Overall - Max    | 36.43% | 33.82% | 35.35% | 35.79% | 38.41%        | 38.75%  | 1.70%  |

---

> > ### Author Response · Authors · 2022-08-08
> > **Another friendly reminder**
> >
> > Dear Reviewer YUN8 for the ReCross paper,
> >
> > Thank you for your time and efforts again in reviewing our paper. We kindly remind you that the discussion period will end very soon. We believe that we sincerely and successfully address your comments, with the results of the supporting experiments.
> >
> > If you have any further concerns or questions, please do not hesitate to let us know.
> >
> > We understand that you are very busy, so we would appreciate it a lot! Thank you very much! :D
> >
> > Sincerely,
> >
> > Authors of the ReCross paper

---

### Author Response · Authors · 2022-08-02
**Summary of our rebuttal and revision**

Dear reviewers,


Thank you for your thoughtful and positive reviews! We are happy to hear that you liked our contributions to unsupervised cross-task generalization. We appreciate all of you for your __positive comments__ highlighting the strengths of our work for a summary:
* __YUN8__: a simple but effective method, impressively works, reasonable set of baselines and an informative choice of ablations; contribution of the reranker; adequately addressed the limitations.
* __fTmN__: “the entire setup being explored is an important and high-impact problem to study”; “a compelling solution worth exploring”; quite interesting result;
* __NtmR__: clearly written, lucidly presented, sound method, non-trivial performance gains


We also sincerely thank reviewers for your constructive feedback and questions to improve our manuscript. We have __addressed all the questions__ raised by reviewers with new experiments during this rebuttal period.


We summarize how we address the main questions as follows:
* __YUN8__: We added the empirical results using the standard EM metric in Appendix D.1. We revised the paragraphs in the main paper about SoftEM for more clarifications.
* __fTmN__: In our response below, we summarized our analysis to better understand the performance gain and model behavior in Appendix (A~B).
* __fTmN__: We used a detailed paragraph to illustrate the importance of the reranker by showing the relative improvement over the ReCross without reranking.
* __fTmN__: We rephrased a few paragraphs to make them more concise and refined Figures 1 and 2 for better visualization.
* __NtmR__: We discussed the cost-effectiveness of ReCross in the unsupervised setting from multiple perspectives in our response below.
* __NtmR__: We added the result comparison under the few-shot settings in Appendix D.2. and found that ReCross can also boost the performance of few-shot learning for task generalization. This finding shows that our approach improves performance even in settings where few-shot data is available or easy to generate.


We submitted our __revised draft and supplementary file__ (i.e., Appendix) that addressed individual concerns. We marked changed parts with blue fonts. To make it more convenient for reviewers to check our appendix, we only uploaded the appendix pdf file as the supplementary file, and our previous code zip is still accessible in the revision history.


Thank you for your consideration,

Authors

---

### Meta-Review · Area_Chair_S4n7 · 2022-08-24

**Recommendation:** Accept
**Confidence:** Certain

**Metareview:**

This paper presents an approach called ReCross that improves zero-shot task performance by retrieving and fine-tuning on examples of similar supervised tasks. This method is shown to help multi-task finetuned models when evaluated zero-shot on novel tasks.

The interesting finding of the paper is that fine-tuning on relevant examples from different but possibly related tasks can help. This finding can help researchers in the areas of zero-shot learning and multitask models.

Otherwise, the method although conceptually simple, includes significant additional machinery, which likely makes it practically difficult to use as the reviewers point out. Similarly, the relative contribution of the re-ranking step seems small and the steps appears to add significant complexity. As one of the reviewers points out, the paper and the method may be clearer without that step.

The review process included a lengthy and productive discussion, which helped the paper clarify and improve on several points. As result two of the reviewers increased their scores. There is now consensus among the three reviewers that the paper should be accepted.



**Award:**

No

---

### Decision · Program_Chairs · 2022-09-14

Accept